# Fuel Properties of Torrefied Biomass from Pruning of Oxytree

**Kacper Świechowski [1], Marek Liszewski [2], Przemysław Bąbelewski [3], Jacek A. Koziel [4] and Andrzej Białowiec [1,\*]**

[1] Faculty of Life Sciences and Technology, Institute of Agricultural Engineering, Wrocław University of Environmental and Life Sciences, 37/41 Chełmońskiego Str., 51-630 Wrocław, Poland; kacper.swiechowski@upwr.edu.pl

[2] Faculty of Life Sciences and Technology, Institute of Agroecology and Plant Production, Wrocław University of Environmental and Life Sciences, 24A Grunwaldzki Sq., 53-363 Wrocław, Poland; marek.liszewski@upwr.edu.pl

[3] Faculty of Life Sciences and Technology, Department of Horticulture, Wrocław University of Environmental and Life Sciences, 24A Grunwaldzki Sq., 53-363 Wrocław, Poland; przemyslaw.babelewski@upwr.edu.pl

[4] Department of Agricultural and Biosystems Engineering, Iowa State University, Ames, IA 50011, USA; koziel@iastate.edu

**\*** Correspondence: andrzej.bialowiec@upwr.edu.pl; Tel.: +48-71-320-5973

**Abstract:** The very fast growing Oxytree (*Paulownia Clon in Vitro 112*) is marketed as a promising new energy crop. The tree has characteristically large leaves, thrives in warmer climates, and requires initial pruning for enhanced biomass production in later years. We explored valorizing the waste biomass of initial (first year) pruning via thermal treatment. Specifically, we used torrefaction ('roasting') to produce biochar with improved fuel properties. Here for the first time, we examined and summarized the fuel properties data of raw biomass of Oxytree pruning and biochars generated via torrefaction. The effects of torrefaction temperature (200~300 °C), process time (20~60 min), soil type, and agro-technical cultivation practices (geotextile and drip irrigation) on fuel properties of the resulting biochars were summarized. The dataset contains results of thermogravimetric analysis (TGA) as well as proximate and ultimate analyses of Oxytree biomass and generated biochars. The presented data are useful in determining Oxytree torrefaction reaction kinetics and further techno-economical modeling of the feasibility of Oxytree valorization via torrefaction. Oxytree torrefaction could be exploited as part of valorization resulting from a synergy between a high yield crop with the efficient production of high-quality renewable fuel.

**Dataset:** This dataset is available in the supplementary files.

**Dataset License:** CC-BY

**Keywords:** renewable energy; biochar; biomass valorization; torrefaction; wood; fuel properties; paulownia; proximate analysis; ultimate analysis; Oxytree; carbon sequestration

---

## 1. Summary

The global energy consumption increased by ~80% between 1980 and 2010. It is predicted that energy demand will increase by another 50% by 2025. New solutions for energy production and diversifying sources in addition to conventional fossil fuels are needed [1,2]. The environmental impact of fuel combustion (e.g., air pollution) and the sequestration of $CO_2$ need to be considered [2,3]. One approach is to replace conventional fuel with biofuel as a means to limit overall $CO_2$ emissions [2,3].

For example, mixing gasoline with bioethanol [3] and co-combustion coal with biomass [4] are used. Growing energy crops that sequester $CO_2$ can also be considered as an alternative to conventional fuels. Paulownia trees could be a basis for cleaner technology development.

One species of the genus *Paulownia* is *Clon in Vitro 112* (also known as Oxytree), a new type of energy crop gaining popularity in Europe. The genus *Paulownia* originates from southeastern China and Japan. Paulownia species have been introduced to the United States [5]. In recent years, these trees have become more popular for timber purposes in warmer climates [6], and are cultivated in southern Europe [7]. Oxytree is a hybrid of *Paulownia elongata* and *Paulownia fortunei*, grown in vitro. The Clon 112 was added to the Plant Variety Office register (EU) in 2011 [8]. In Poland, the first experimental plantations were established in 2016. Since then, 600 ha of plantations have been created. Oxytree is a very fast-growing plant with the C4 photosynthesis pathway [9].

The distinguishing feature of Oxytree compared to other energy crops is the ability to produce high-quality wood. This wood can be used for industrial purposes, such as the production of particleboard. Oxytree wood has a high strength-to-mass ratio [7]. The modulus of rupture and modulus of elasticity are 29 MPa and 3970 MPa, respectively [10]. However, Paulownia species have a very soft wood (10–25 MPa on Janka's scale) [11].

The density of Paulownia plantings determines the utilization of biomass. In the case of wood for industrial harvest, planting should not exceed 600 tree·ha$^{-1}$. However, it could reach 3000 tree·ha$^{-1}$ for energetic purposes [12]. The higher heating value (HHV) for biomass in the Paulownia genus differs slightly depending on the variety and ranges from 19.2~19.6 MJ·kg$^{-1}$ [10]. The HHV of *Paulownia tomentosa* is 19.7 MJ·kg$^{-1}$ [13] and is comparable with willow and poplar biomass HHV, ranging from 18.5~19.9 and from 19.6~19.8 MJ·kg$^{-1}$, respectively [14]. Short-rotation woody crop (SRWC) plantations are developed in Europe and they include poplar, black locust, willow [15], and Paulownia trees. Wood from these plantations may be utilized for the production of heat and electricity, as well as the production of bioethanol [16]. Torrefaction is considered a novel approach for biomass valorization [16]. Thus, we are pioneering the potential use of Paulownia valorization via torrefaction.

Torrefaction is the thermochemical process (similar to 'roasting')—typically at temperatures between 200 °C and 300 °C—occurring in the absence of oxygen, resulting in biochars [17]. Torrefaction allows the conversion of raw biomass into solid fuel with energy properties similar to bitumen and lignite coal [18], depending on the initial properties of the feedstock. In addition, the torrefied biomass storage and transportation conditions are improved. Produced biochars are hydrophobic and have increased energy density due to volume reduction. Torrefaction can convert food waste, sewage sludge, a combustible fraction of municipal solid waste (also known as refuse-derived fuel, RDF) [19–21], high moisture herbaceous residues [22], and forest residues [23,24] into fuel.

To date, the torrefaction of Oxytree has not been attempted yet. This dataset includes the pioneering work results covering thermogravimetric analyses (TGA), proximate and ultimate properties of Oxytree, and generated biochars. Oxytrees were grown on two types of soils in Poland with the application of four types of cultivation. Plants were pruned after the first year of vegetation. Pruned biomass created an early opportunity to investigate the fuel properties of Oxytree. The presented data are useful in determining the Oxytree torrefaction reaction kinetics and the techno-economical modeling of the feasibility of Oxytree valorization via torrefaction, including a comparison with other torrefied energy crops.

## 2. Data Description

### 2.1. The Origin of Biomass

The samples of pruned biomass (as presented in Figures 1 and 2) of *Paulownia Clon in Vitro 112* originated from two experimental plantations with different soils. The plantation in Pawłowice (Wrocław, Poland) was established on sandy soil, classified as V soil class belonging to brunic arenosols,

and the plantation in Psary (Wrocław, Poland) was established on clay soil classified as Phaeozems. Classes of soils refer to the world reference database of soil resources [25].

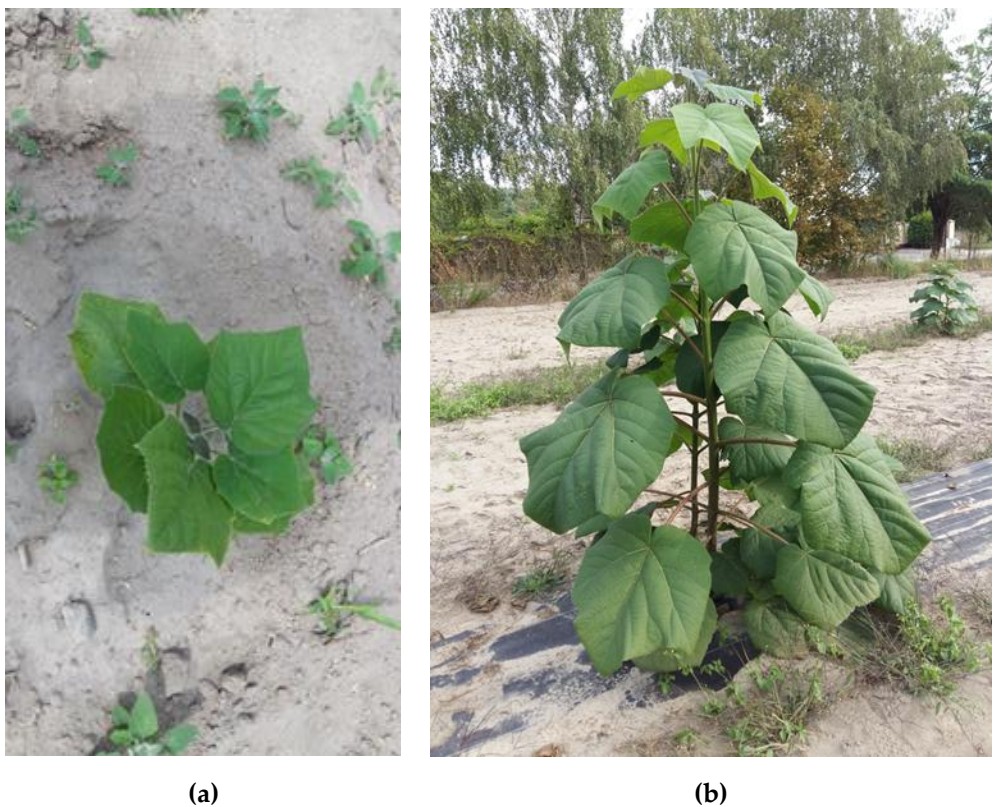

**(a)**                                                                 **(b)**

**Figure 1.** Paulownia tree before pruning. Planted on 16 May 2016. Image (**a**) was photographed on 13 June. Image (**b**) was photographed on 27 September 2016; the Oxytree was approximately 0.9 m tall. Geotextile is visible in the front row (as one of the treatments) in image (**b**). Oxytrees in the background row were cultivated without geotextile.

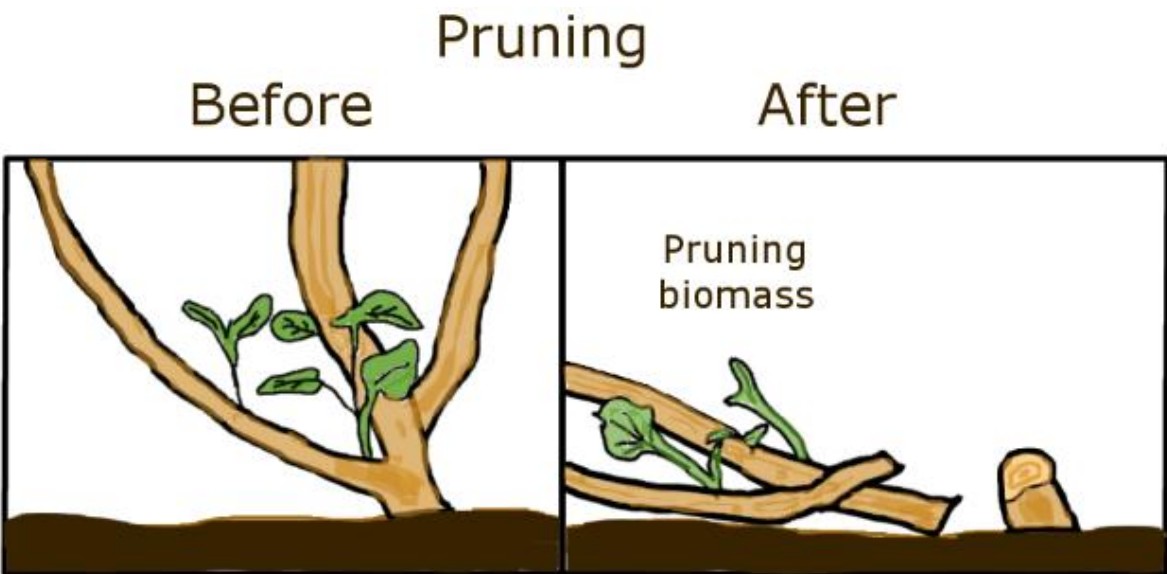

**Figure 2.** Schematic of Oxytree biomass pruning was on 27 September 2016, i.e., approximately 5 months after planting. Pruned biomass was torrefied and analyzed for fuel properties.

In vitro shoots propagated to ~20–40 cm were planted with 4 m × 4 m (16 m² per tree) spacing on 19 May 2016. The trees subjected to analysis were pruned on 27 September 2016, as is presented in Figure 2. Analyzed trees represent biomass after 1 year of vegetation. Normal agro-technical care treatment takes place 12 months after planting in May. This agro-technical care treatment involves trimming 0.05 m above the pitch to derive one main shoot, which would become the main trunk of the tree.

At both plantations, the mineral fertilization was applied as follows: pre-sowing fertilization with a dose of 70 kg $K_2O \cdot ha^{-1}$ (in the form of 60% potassium salt), 40 kg $P_2O_4 \cdot ha^{-1}$ (in the form of 46% triple superphosphate), and 40 kg·ha$^{-1}$ of nitrogen (in the form of 46% urea). Additionally, nitrogen with doses of 20 kg·ha$^{-1}$ (in the form of ammonium nitrate) was supplied monthly.

At the initial stage of the plants' development, 1 month after planting, all plants were irrigated (50 dm³·tree$^{-1}$ per month, irrigated twice a week). Management of weeds included mechanical operations. Weeds in the inter-row were removed mechanically with a weeder and an active harrow. Weeds were removed mechanically every 2 months. Weeds between the plants in a row were removed manually using a hook. The schematic of planting on each plantation is presented in Figure 3.

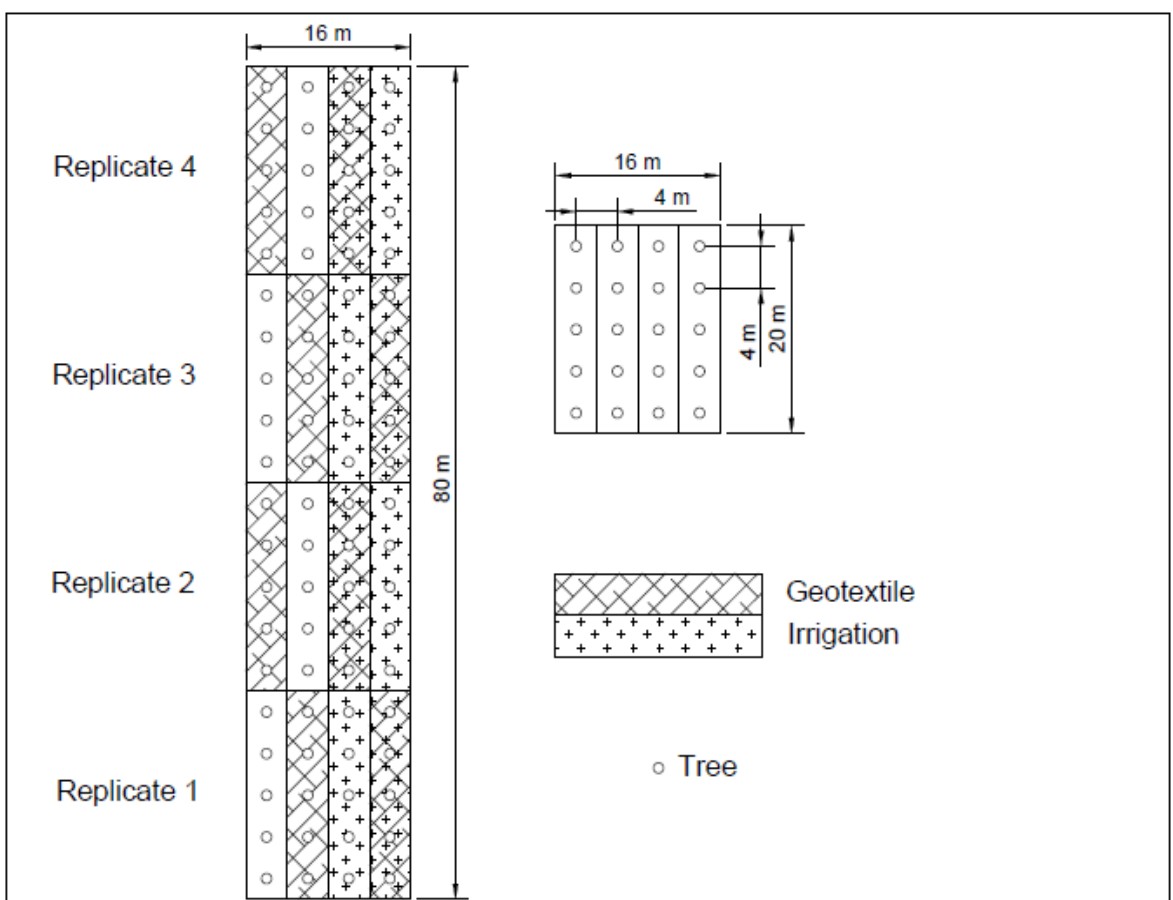

**Figure 3.** Aboveground schematic of Oxytree planting and experimental design. The same design was used on two experimental plantations, one with clay and and one with sandy soils.

The experiment differed on the applied cultivation method. The first differentiating factor was the application of a geotextile. Geotextile is a kind of material used to cover the ground under plants. It is used to provide protection against weeds. Geotextile also decreases water evaporation from the ground and increases the temperature of the ground [26]. A black geotextile was used (Wigolen S.A., model TA 94, Częstochowa, Poland). The second differentiating factor was the application of additional irrigation. Five trees in the plot were supplied with water by drip irrigation without compensation.

Each plot was irrigated for 2 h twice a week with a hydraulic loading rate of 312.5 dm$^3$·h$^{-1}$. The area of each irrigated plot was ~80 m$^2$. The matrix of the experiment is given in Table 1. The temperature conditions during the growing season from May to October were 15.3 °C, 18.6 °C, 19.5 °C, 17.9 °C, 16.4 °C, and 8.5 °C, respectively. However, precipitation was somewhat abnormal, i.e., 5 mm in May (typically 54 mm), 114 mm in July (typically 78 mm), 27 mm in August (typically 65 mm), and 84 mm in October (typically 34 mm).

**Table 1.** The matrix of the experiment involving the effects of geotextile, drip irrigation, and the soil type of Oxytree pruned biomass properties.

| Cultivation Type Symbol | Application of Geotextile | Application of Irrigation | Soil Type |
|---|---|---|---|
| S(G-)(I-) | - | - | S—sandy soil, classified as V soil class belonging to brunic arenosols [21] |
| S(G+)(I-) | + | - | |
| S(G-)(I+) | - | + | |
| S(G+)(I+) | + | + | |
| C(G+)(I-) | + | - | C—clay soil classified as Phaeozems [21] |
| C(G-)(I-) | - | - | |
| C(G+)(I+) | + | + | |
| C(G-)(I+) | - | + | |

The biomass samples were collected at the end of the first vegetation season on 27 September 2016. The fresh mass yield of biomass was ~0.49 Mg·ha$^{-1}$ (assuming 625 trees per hectare). Detailed information about mass yield is given in the Supplementary Material file "Oxytree torrefaction data.xlsx" in the spreadsheet "Oxytree biomass yield". The (mass-based) content of leaves and shoots in the biomass samples was about 71% and 29% of the total fresh mass, respectively. In dry mass, the content of leaves was about 66% and that of shoots was about 34%. Oxytree biomass samples were marked according to symbols used in Table 1.

*2.2. Properties of Raw and Torrefied Biomass*

The thermogravimetric analysis (TGA) of the torrefaction process was conducted with each kind of Oxytree biomass according to the protocol described elsewhere [27]. In brief, torrefied Oxytree biomass (biochar) was produced from each Oxytree biomass cultivation variant. The biochars were generated under a temperature of 200~300 °C (with 20 °C intervals) and retention times of 20~60 min (with 20 min intervals). The obtained data are presented in the Supplementary Material file "Oxytree torrefaction data.xlsx" in five sheets:

- Read me (guide);
- Oxytree biomass yield;
- Oxytree torrefaction TGA ;
- Proximate analyses of Oxytree biomass and biochars;
- Ultimate analyses of Oxytree biomass and biochars.

The first "Read me" sheet is a guide on how to read the data with short information about each type of treatment. The second spreadsheet ("Oxytree biomass yield") contains data on the Oxytree biomass yield, energy densification ratio, mass, and energy yield of biochars. The third spreadsheet ("Oxytree torrefaction TGA") contains raw data from TGA tests. The fourth spreadsheet ("Proximate analyses") contains data on moisture content, organic matter content, lost on ignition content, and ash content in raw and torrefied Oxytree biomass. The fifth spreadsheet ("Ultimate analyses") presents the elemental composition, H:C and O:C ratios of raw Oxytree biomass and biochars, high heating value (HHV), low heating value (LHV), and HHV without ash.

## 3. Methods

Collected Oxytree biomass samples were dried for 24 h in a WAMED laboratory dryer, model KBC-65W (Warsaw, Poland), under the temperature of 105 °C. Then, dry biomass samples were ground through a 1 mm screen. For biomass grinding, the laboratory knife mill TESTCHEM, model LMN-100 (Pszów, Poland), was used.

### 3.1. Torrefaction Process

The torrefaction was carried out in a muffle furnace (Snol 8.1/1100, Utena, Lithuania). $CO_2$ was used as an inert gas with a flow of 10 $dm^3 \cdot h^{-1}$. The process was carried out under the setpoint temperatures of 200 °C to 300 °C with intervals of 20 °C. For each temperature, torrefaction was carried out in 20, 40, and 60 min. The samples were heated from 20 °C to the torrefaction setpoint temperature with a heating rate of 50 $°C \cdot min^{-1}$. The dry mass of the sample used for the tests was 10 g (± 0.5 g). The biochars were removed from the muffle furnace when the interior temperature was lower than 200 °C. The approximate times of cooling from 300 °C, 280 °C, 260 °C, 240 °C, and 220 °C to 200 °C were 38, 33, 29, 23, and 13.5 min, respectively. The approximate cooling time from 300 °C to room temperature (~20 °C) was around 6 h. Analyses of one replicate were carried out. The TGA analysis was conducted in three replicates. The analyzer is described elsewhere [23].

Mass yield, energy densification ratio, and energy yield were calculated based on Equations (1)–(3) [28].

$$Mass\ yield = \frac{Mass\ of\ dry\ biochar}{Mass\ of\ dried\ raw\ material} \cdot 100 \tag{1}$$

$$Energy\ densification\ ratio = \frac{HHV\ of\ biochar}{HHV\ of\ raw\ material} \tag{2}$$

$$Energy\ yield = mass\ yield \cdot energy\ densification\ ratio \tag{3}$$

where:

Mass of dry biochar—mass of (dry) biochar after the process of torrefaction, g;

Mass of dried raw material—dried mass of biomass used in the process of torrefaction, g;

100—conversion to percent;

HHV of biochar—high heating value of biochar after the process of torrefaction, $J \cdot g^{-1}$;

HHV of raw material—high heating value of dried biomass (raw material) used for torrefaction, $J \cdot g^{-1}$.

### 3.2. Proximate Analysis

Raw and torrefied biomass samples were tested in three replicates for:

- Moisture content, determined in accordance with [29], by means of the laboratory dryer (WAMED, model KBC-65W, Warsaw, Poland).
- Organic matter, determined in accordance with [30], by means of the SNOL 8.1/1100 muffle furnace (Utena, Lithuania). The dried sample was heated at 550 °C for at least 1 h (1 g and 3 h in this study) in the muffle furnace. After that, a sample was cooled down in a desiccator to room temperature. Next, the mass values before and after the process (measured with an accuracy fo ±1 mg) were used to calculate the loss on ignition. Organic matter content was estimated as the initial mass of the dry sample minus loss of ignition.
- Combustibles and ash content, determined in accordance with [31], by means of the SNOL 8.1/1100 muffle furnace (Utena, Lithuania). This method consists of incinerating the sample and then calcining it to a constant mass at a temperature of 815 ± 10 °C for 2 h. The last step is a calculation in percent of the content of the combustibles and ash content.
- High heating value (HHV) and low heating value (LHV), determined in accordance with [32], by means of the IKA C2000 Basic calorimeter. HHV was determined by the calorimeter and

the mass of dry samples was 0.3 g. Based on HHV, moisture content, and the ultimate analysis, the LHV from Equation (4) was determined.

$$Q_i = Q_s r \cdot (W + 8.94 \cdot H) \tag{4}$$

where:

Qi—low heating value, J·g⁻¹;
Qs—high heating value, J·g⁻¹;
R—the heat of water evaporation, 22.42 J·g⁻¹, for 1% of the water in fuel;
W—moisture content, %;
H—hydrogen content, %;
8.94—hydrogen to water conversion, -.

- HHV$_{daf}$ (on dry and on ash-free bases) was calculated according to Equation (5) [33]:

$$HHV_{daf} = \frac{HHV}{M_f - M_{ash}} \tag{5}$$

where:

HHV$_{daf}$—high heating value on dry and ash-free bases, J·g⁻¹;
HHV—high heating value (on dry basis), J·g⁻¹;
M$_f$—fuel mass (on dry basis), mass, g;
M$_{ash}$—mass of ash in fuel, g.

### 3.3. Ultimate Analysis

Carbon, hydrogen, and nitrogen contents were determined by means of the elemental CHNS analyzer CE Instruments Ltd (Manchester, UK). Sulfur was determined by the atomic emission spectrometry method with excitation in inductively coupled plasma (ICP-AES) after microwave mineralization, Thermo Fisher Scientific, iCAP 7400 ICP-OES (Waltham, USA). One replicate was carried out.

The oxygen was determined by the calculation method according to Equation (6):

$$O = 100\% - C - H - N - S - Ash \tag{6}$$

where:

O—oxygen content, %;
C—carbon content, %;
H—hydrogen content, %;
N—nitrogen content, %;
S—sulfur content, %;
Ash—ash content, %.

Atomic ratios for H:C and O:C were determined using Equations (7) and (8) [34]:

$$H : C = \frac{\frac{H}{1}}{\frac{C}{12}} \tag{7}$$

$$O : C = \frac{\frac{O}{16}}{\frac{C}{12}} \tag{8}$$

where:

H:C—atomic ratio of H to C, -;

O:C—atomic ratio of O to C, -;

1—atomic mass of H, u;

12—atomic mass of C, u;

16—atomic mass of O, u.

## 4. User Notes

The given dataset presents pioneering work on the Oxytree biomass torrefaction. The dataset consists of results of biomass yield, energy densification ratio, mass, and energy yield of biochars in relation to the cultivation regime and soil type. The torrefaction TGA analyses results were given for pruned Oxytree biomass. The fuel characterization of raw Oxytree biomass and produced biochars were shown. These data can be used for calculating Oxytree torrefaction kinetics and activation energy, and can also be used to propose mathematical models describing the influence of torrefaction, temperature, and residence time on fuel properties. The presented data may serve as reference values for comparative analyses with other types of energy crops biomass such as poplar, willow, eucalyptus, *Robinia*, and birch.

**Supplementary Materials:** The following are available online at http://www.mdpi.com/2306-5729/4/2/55/s1, File S1: Oxytree torrefaction data.xlsx.zip.

**Author Contributions:** Conceptualization, A.B.; methodology, A.B., M.L. and P.B.; software, K.Ś.; validation, K.Ś., A.B. and J.A.K.; formal analysis, K.Ś.; investigation, K.Ś., M.L. and P.B.; resources, K.Ś., M.L, P.B., A.B. and J.A.K.; data curation, K.Ś.; writing—original draft preparation, K.Ś.; writing—review and editing, K.Ś., A.B. and J.A.K.; visualization, K.Ś.; supervision, A.B. and J.A.K.; project administration, A.B.; funding acquisition, K.Ś., A.B. and J.A.K.

**Funding:** "The PROM Programme—International scholarship exchange of Ph.D. candidates and academic staff" is co-financed by the European Social Fund under the Knowledge Education Development Operational Programme PPI/PRO/2018/1/00004/U/001. The authors would like to thank the Fulbright Foundation for funding the project titled "Research on pollutants emission from Carbonized Refuse-Derived Fuel into the environment," completed at the Iowa State University. In addition, this paper preparation was partially supported by the Iowa Agriculture and Home Economics Experiment Station, Ames, Iowa. Project no. IOW05556 (Future Challenges in Animal Production Systems: Seeking Solutions through Focused Facilitation) sponsored by Hatch Act and State of Iowa funds.

**Conflicts of Interest:** The authors declare no conflict of interest. The funders had no role in the design of the study; in the collection, analyses, or interpretation of data; in the writing of the manuscript; or in the decision to publish the results.

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
