# Peer review of "Fuel Properties of Torrefied Biomass from Pruning of Oxytree"

_data, 2019_

Round 1

Reviewer 1 Report

This data manuscript presents Paulownia growth and torrefaction information that include material yields, TGA, proximate analysis elemental analysis and heating values. The manuscript contains useful data and can be accepted if the following minor changes are addressed.

Please check again and see if you have mentioned replicates for torrefaction, proximate, ultimate, TGA etc. I can see it for some of them in the excel sheet but not in Methods section.

Fig. 1, 2 captions: Please write year of of plantation

Spelling of “pruning” in Fig. 2

Fig. 3 – what are the circles? Trees?

 “The temperature conditions during the growing season (from May to October) was typical for this geographical area” What were the actual temperature values/range for the growing year?

“The geotextile is a kind of material used to cover the ground under plants. It is used to provide protection against weeds. Geotextile also decreases water evaporation from the ground and increases the temperature of the ground.” citation(s).

How long does it take to reach the temperature setpoints, and cooling to room temperature, in the muffle furnace?

“Then dry biomass samples were ground to a size below 1 mm.” Size is not clear. Do you mean through 1 mm screen using a knife mill?

Does energy density have units?

Initial mass of sample: Did you measure moisture content of biomass after drying? Is this the same as “Moisture%” column in Proximate analysis sheet? MY equation: Is it initial mass of sample on dry basis. or should it be initial mass of sample*solids content?

What does the abbreviation “daf” mean?

CHNOS%: Is this on a dry basis corrected for moisture in sample?

ED equation 3: What is MY prime dash?

You may consider briefly expanding on some of the methods like combustibles, ash, organic content etc. as the standards/references can be hard to find and the standards seem to be only available in a language other than English.

Author Response

We included our responses to reviewer's comments in the attached file.

Reviewer 2 Report

In “Introduction” section, a more complete literature review should be conducted about the energy consumption from industrial and transportation sectors, which is a world problem for city governance. The world’s energy demand, in fact, is nowadays growing more and more and the problem of the fossil fuels depletion is becoming increasingly crucial. It is also important remember that currently more than 25 billion tons of CO2 arising from worldwide human activities are released annually into the atmosphere. For this reason the development of new technologies in energy consumption management and the changing from conventional fuel to biofuel and biomass are stringent necessity, both to meet the energy demands and to limit the production of carbon dioxides. In order to add all these considerations, you may consider in the references these two pertinent papers:

P. Iodice, A. Senatore. Industrial and Urban Sources in Campania, Italy: The Air Pollution Emission Inventory. Energy & Environment, 26(8), 2015, pp. 1305–1318

P. Iodice, A. Senatore, G. Langella, A. Amoresano. Advantages of ethanol–gasoline blends as fuel substitute for last generation Si engines. Environmental Progress and Sustainable Energy, 36, 4, 2017, 1173-1179

Figure 3 must be better described

Line 212: Authors affirm that presented data may serve as reference values for comparative analyses with other types of energy crops biomass. This consideration should be better explained

Author Response

(The authors gave the same response as above.)

Reviewer 3 Report

The study entitled Fuel properties of torrefied biomass from pruning of Oxytree valorized the waste biomass of first year pruning via thermal treatment, with the use of torrefaction to produce biochar with improved fuel properties. In this study the fuel properties data of raw biomass of Oxytree pruning and biochars generated via torrefaction are summarized.

The scientific data provided is comprehensive and can be further employed for numerical assessment and analysis. The subject is interesting for the readership of Data and the material provided is adequate. The structure and language of the paper is appropriate, and the study can be accepted for publication.

Author Response

(The authors gave the same response as above.)
